# Wild Rats, Laboratory Rats, Pet Rats: Global Seoul Hantavirus Disease Revisited

**DOI:** 10.3390/v11070652

**Published:** 2019-07-17

**Authors:** Jan Clement, James W. LeDuc, Graham Lloyd, Jean-Marc Reynes, Lorraine McElhinney, Marc Van Ranst, Ho-Wang Lee

**Affiliations:** 1National Reference Centre for Hantavirus Infections, Laboratory of Clinical and Epidemiological Virology, University Hospital of Leuven, 3000 Leuven, Belgium; 2Galveston National Laboratory, University of Texas Medical Branch, Galveston, TX 77555-0610, USA; 3Laboratory for Public Health England, Porton Down, Wiltshire SP4 OJG, UK; 4National Reference Centre for Hantaviruses, Unité Environnement et Risques Infectieux, Institut Pasteur, 75724 Paris CEDEX 15, France; 5Wildlife Zoonoses and Vector Borne Diseases Group, Animal and Plant Health Agency, Surrey KT15 3NB, UK; 6WHO Collaborating Centre for Hemorrhagic Fever with renal Syndrome, National Academy of Science, Seoul 06579, Korea

**Keywords:** hantavirus, Seoul virus (SEOV), brown rat, wild rat, laboratory rat, pet rat, hemorrhagic fever with renal syndrome (HFRS), acute kidney injury (AKI), hantavirus cardio-pulmonary syndrome (HCPS), hantavirus disease

## Abstract

Recent reports from Europe and the USA described Seoul orthohantavirus infection in pet rats and their breeders/owners, suggesting the potential emergence of a “new” public health problem. Wild and laboratory rat-induced Seoul infections have, however, been described since the early eighties, due to the omnipresence of the rodent reservoir, the brown rat *Rattus norvegicus*. Recent studies showed no fundamental differences between the pathogenicity and phylogeny of pet rat-induced Seoul orthohantaviruses and their formerly described wild or laboratory rat counterparts. The paucity of diagnosed Seoul virus-induced disease in the West is in striking contrast to the thousands of cases recorded since the 1980s in the Far East, particularly in China. This review of four continents (Asia, Europe, America, and Africa) puts this “emerging infection” into a historical perspective, concluding there is an urgent need for greater medical awareness of Seoul virus-induced human pathology in many parts of the world. Given the mostly milder and atypical clinical presentation, sometimes even with preserved normal kidney function, the importance of simple but repeated urine examination is stressed, since initial but transient proteinuria and microhematuria are rarely lacking.

**Highlights:** Rat (wild, laboratory, or pet)-induced hantavirus disease is one of the oldest known forms of hantavirus infection, and the only one with a global spread, due to the omnipresence of its rodent reservoir. The causative agent, Seoul orthohantavirus (SEOV), is moreover the only orthohantavirus species with multiple isolations in four continents, mostly made in the pioneer early 1980s. Paradoxically, SEOV-induced hantavirus disease is apparently still often missed or misdiagnosed, perhaps due to its often mild and atypical clinical presentation [1]. 

## 1. Introduction

Hantaviruses are the cause of “emerging” rodent-borne viral hemorrhagic fever (VHF), known to western medicine since the early 1980s. They are mostly linked to a unique species of asymptomatic rodent reservoirs with a distinct, but geographically well-limited biotope, shedding infectious viral particles in their excreta. Transmission to humans occurs mostly via the inhalation of these particles, causing multiple forms of organ pathology, via a transient so-called “pro-inflammatory cytokine storm”. Old World hantaviruses mainly target the human kidney, causing “hemorrhagic fever with renal syndrome” (HFRS), whereas New World hantaviruses, discovered since 1993, mainly target the lung, causing “hantavirus cardio-pulmonary syndrome” (HCPS), but many clinical overlaps between HFRS and HCPS are increasingly reported. Rather paradoxically, the brown rat (*Rattus norvegicus)*, being the only global hantavirus rodent reservoir, clearly also present in the New World, deserved little scientific interest in past decades, thus maintaining an original misconception that HFRS was a disease representative for the Old World, and HCPS for the New World [1].

The recent discovery, both in Europe and the USA, that SEOV-infected pet rats could also cause HFRS in their owners or breeders resulted in a sudden resurge of the research into rat-transmitted hantaviral disease.

## 2. Prelude

In December 2016, an owner of a Wisconsin (USA) in-home rattery was hospitalized with fever, leukopenia, elevated transaminases and proteinuria. Although no typical acute kidney injury (AKI) was reported, the patient was serologically diagnosed with an acute Seoul orthohantavirus (SEOV) infection. However, mild and atypical clinical presentations are a constant feature of SEOV-induced HFRS forms, as repeatedly shown hereunder [1]. A family member later developed similar symptoms, but was not hospitalized [2]. In late December, 2016, a report of mild AKI followed in an 18-year-old female from Tennessee (USA) (hematuria and peak serum creatinine level of 1.27 mg/dL). In April 2017, a second case was reported in her 38-year old mother who was hospitalized for gastro-intestinal symptoms with dyspnea and even milder AKI (peak serum creatinine 1.13 mg/dL, with mild thrombocytopenia of 143,000/µL). In the latter case, SEOV RNA was detected by reverse transcription-polymerase chain reaction (RT-PCR). Both cases appeared to have been caused by pet rat-transmitted SEOV, and both recovered without treatment or complications [3,4]. The USA Center of Disease Control and Prevention (CDC) launched a nationwide investigation, identifying 31 rat-breeding facilities with human and/or rat SEOV infection in 11 USA states, six of which exchanged rats with Canadian ratteries. A total of 24/183 (13.1%) US or Canadian residents associated with these facilities were found to be serologically SEOV-positive, resulting in three hospitalizations in the USA. No AKI was mentioned, and spontaneous recovery occurred, whereas no serious illness was reported from Canada [2]. Similar cases, with or without AKI, and all induced by apparently healthy pet rats, were described in the UK [5,6] and in France [7]. Recently, increasing reports of wild and pet rat-associated SEOV-induced hantavirus disease in the USA and Europe reflected a renewed awareness of the disease outside of Asia, breaking years of misconceptions about the true distribution and clinical aspects of hantavirus diseases. 

The decades-long assumption that HFRS was limited to the Old World, and HCPS restricted to the Americas, ignored the remarkable global progress made by pioneer hantavirologists in the early 1980s. Rat-transmitted hantavirus or SEOV is the second oldest isolated hantavirus (1980) [8], the first isolated hantaviral pathogen in the USA (1984) [9], and the most widely distributed hantaviral pathogen, due to the omnipresence of its reservoir and vector, the brown or Norway rat (*Rattus norvegicus*). Before the unexpected 1993 discovery of two other autochthonous American hantaviral pathogens, Sin Nombre orthohantavirus (SNV), followed by Andes orthohantavirus (ANDV), early hantavirology in the USA focused primarily on SEOV and its possible impact on human health. A 1994 compilation of the then existing world literature confirmed hantavirus-antibodies, -antigen, and -virus isolation in two different classes of animals (mammals and birds), eight different orders, 24 different families and a total of 164 different species, mostly small mammals. Of these, the rat was by far the species yielding most positive results, i.e., in 34 different countries for the brown rat, 20 countries for the roof rat (*Rattus rattus*), 19 countries for non-specified *Rattus* species (which may include pet rats), and in 13 countries for laboratory rats [10]. To date, SEOV is still the only hantavirus with multiple isolations in four different continents (Asia, Europe, America, and Africa) (see hereunder), a unique accomplishment by different hantavirus researchers beginning 1980s, considering the inherent difficulties associated with isolating hantaviruses. So far, the limited attempts for pet rat isolations of SEOV, both from rats and humans, have been unsuccessful. 

In the Far East, and particularly in China, extensive annual mixed epidemics, caused by prototype Hantaan orthohantavirus (HTNV) and/or SEOV, resulted in tens of thousands of human infections yearly. In contrast, although American authors once called “*the SEOV prevalence in [wild] rats in many United States cities among the highest recorded globally*” [11,12,13], there has been little noticeable impact over the subsequent three decades on human morbidity and/or mortality in North America. Similarly, North-American pet rats have had only limited reported consequences for public health so far [2]. The novel notion, however, that even fancy pet rats might infect their owners or breeders with a zoonotic agent (SEOV), previously thought to be limited to wild rats, has the potential to dramatically change our insights about the global importance of rat-transmitted HFRS, since pet rat trade is international, and likely also to be found beyond Europe and North America [13]. 

Additionally, outbreaks among laboratory workers in Asia, Russia, and Europe in the 1960s and 1970s, associated with SEOV-infected laboratory rats [13], played a pivotal role in understanding the “new” human illness, later called HFRS after a World Health Organization (WHO) meeting, during which the clinical characteristics of HFRS were, be it succinctly, described [14]. This early clinical and epidemiological knowledge was overshadowed a decade later with the discovery of a new and often lethal disease caused by SNV or ANDV, and called HCPS [15]. 

In Section 3, “Overview”, we discuss the recorded history of SEOV [10,16,17,18,19] and SEOV-induced human infections in four continents, followed by the current status of pet and domesticated rat SEOV infections. 

## 3. Overview of Chronological SEOV Findings by Continent

### 3.1. SEOV in South-East Asia and in the Far East (Figure 1, Asia)

Clinical and later serological recognition of a viral malady, unkown to the West, but caused by the omnipresent an often domestical brown rat, was pivotal for the development of hantavirology in the early 1980s, this time outside of the Far East, Korea being a frontrunner, but rapidly followed by Japan, and particularly by China, where the extensive public health problem caused by recurrent waves of SEOV-HFRS up to today, is still underestimated in Western medical literature (Figure 1). 

#### 3.1.1. SEOV in Korea and South-East Asia

##### SEOV in Korea

The first isolation of HTNV in 1976 [23] was from the Korean striped field mouse (*Apodemus agrarius coreae*). HTNV became the prototype of all ensuing hantaviruses, but was rapidly followed by the isolation of another hantavirus in 1980, retrieved from an urban brown rat, captured in an apartment building in Seoul, South Korea [8]. It was called SEOV 80-39, after local urban HFRS outbreaks involving hundreds of cases [8]. Originally, a clear distinction between prototype HTNV 76-118 and prototype SEOV 80-39 was not made, since both pathogens seemed initially to induce a similar HFRS clinical picture in humans, and because both strains heavily cross-reacted serologically, i.e., in the indirect immunofluorescent antibody assay (IFA), commonly used then. Consequently, early SEOV findings in rats and humans were often labelled as “HTNV-like”. Prototype HTNV 76-118 continued to be used for many years as an IFA antigen for screening human populations and/or HFRS-like diagnosis worldwide, including regions where no human hantavirus infections had previously been noted. IFA cross-reactions, however, helped pioneer diagnosis in Russian and European HFRS cases, caused in fact by arvicolid orthohantavirus Puumala (PUUV), but it became clear that “HTNV-like” IFA results in some regions were often found in ill, or even asymptomatic, individuals with a history of overt exposure to wild or laboratory rats. In retrospect, some of these early findings may have been due to SEOV infections.

Interest in SEOV gained momentum after a series of outbreaks occurred in several Far-East research laboratories, all causing a mild HFRS-like illness, and occurring among doctors and technicians working with Fisher or Wistar rats. In South Korea, nine laboratory staff and two casual visitors developed HFRS between 1971 and 1979 [24]. Since prototype HTNV was only isolated in 1976, it seems likely that laboratory rat-induced SEOV, rather than *A. agrarius*-borne HTNV, may have been the cause of these early laboratory outbreaks. 

The propagation of HTNV, formerly called “Korean hemorrhagic fever” virus (KHFV), in Fisher or Wistar rats and use of their lung tissue as an IFA antigen was published in 1981 as a convenient way to make the then novel diagnostic IFA technique globally available [25]. Previously, the lungs of *A. agrarius* were used as a source of IFA antigen and availability was restricted primarily to the Far East. At that pioneer time, laboratory rats were considered “an animal model free of wild rodent viruses” [25]. However, two SEOV strains were isolated in 1984 from Korean laboratory rats, KSNUSD 84/30 and 84/34 ([26], p.807). Moreover, unapparent aerosol transmission of “HTNV-like” viruses to laboratory rats from other infected cage mates was shown to be a distinct possibility [27]. Shortly after IFA, another screening technique was developed, and called plaque reduction neutralization tests (PRNT), being a more cumbersome serological test, necessitating live virus cultures, but less prone than IFA to cross-reactions. Still later, a much more reliable molecular genotyping technique was being employed, called reverse transcription-polymerase chain reaction (RT-PCR), now considered as the gold standard. In a first retrospective 1994 study, PRNT-confirmed SEOV infection appeared to be the cause of 25% of HFRS cases in Korea, and showed a milder renal and hemorrhagic involvement than in HTNV-caused cases [28]. 

##### SEOV in South-East Asia

From the 1980s on, it became clear that HFRS in non-endemic areas might be milder, and with “atypical” symptoms involving the liver much more than the kidney. In South-East Asian countries, this often led to missed or incorrect (“viral hepatitis”) clinical diagnoses, instead of HFRS. All these countries had, and still have, the wild rat as the most important reservoir for pathogenic hantaviruses. In Malaysia, a “HFRS case involving the liver” was reported in 1987 [29], followed by “hepatitis-like HFRS” outbreaks involving infected wild and laboratory rat colonies [30]. In Singapore in 1988, 44% (143/329) of laboratory rats were found IFA-seropositive to HTNV, but none had hantaviral antigen in lung tissues. Two of 74 laboratory personnel were also seropositive, but neither had a history of HFRS-like clinical illness [31]. In a subsequent Singapore study, *R. norvegicus* was the predominant species captured, with the highest “HTNV-like” seropositivity (36/113 or 32%), resulting in the isolation of R36, a SEOV strain. Two Singapore patients had marked liver dysfunction, but only mild renal involvement, confirming the concept of a “new hepatitis-like hantaviral disease” [32]. Much later, RT-PCR and sequencing of samples from different captured rat species in Singapore resulted in the characterization of SEOV strains Singapore RN 41 and 46 from the brown rat, and strain Jurong TJK/06 from the Oriental house rat, *R. tanezumi* [33]. The latter strain, with Serang virus from the Oriental house rat in Indonesia, Anjozorobe virus from the roof rat in Madagascar, and Mayotte virus from the roof rat in Mayotte Island are variants of the species *Thailand orthohantavirus* (THAIV), formerly (1985) isolated from greater bandicoot rats (*Bandicota indica*) in Thailand [34,35,36,37]. Already in 1994, it was clear that murid THAIV showed marked serological cross-reactions with SEOV strains [38]. Brown rats with Singapore RN 41 and/or 46 were found all over the island, suggesting they were not a recent introduction. Nevertheless, the authors noted the striking local paucity of typical HFRS, with only one sero-confirmed case in Singapore for the last 15 years [33]. 

In 1985, four SEOV isolations were made from urban rats in Hong Kong (strains R 85/14, 19, 35, and 40) [39]. Rat-associated hantavirus infection in both rat and human sera in Hong Kong was subsequently confirmed by IFA and PRNT [40], and two HFRS cases were serologically confirmed ([26], pp. 804 and 807). In rural and urban areas of Cambodia, anti-HTNV IgG was found in 6.4% (13/203) of roof rats, 20.9% (39/187) of brown rats, and none in 183 Pacific rats (*R. exulans*) or in 75 greater bandicoot rats. In 87% of seropositive rodents, RT-PCR confirmed the presence of SEOV RNA from brown rats and of THAIV from the roof rat, [41,42]. To our knowledge, however, no HFRS case has been reported from Cambodia. SEOV was also detected from wild brown rats in Vietnam [43,44,45,46,47] and four HFRS cases were attributed to SEOV using serological tests [43,44]. Finally, among 115 cases hospitalized in Thailand for fever of unknown origin (FUO), one appeared IgM- and IgG-positive by HTNV ELISA and IgG-positive by HTNV IFA, showing acute encephalitis, thrombocytopenia, coagulation anomalies, and elevated liver transaminases, now commonly called “transaminitis” [48]. From the same country, 1/260 leptospirosis-suspected patients was found seronegative for leptospirosis, but had high-titers by HTNV IFA and by PRNT for THAIV, suggesting a pathogenic role in Indochina for THAIV [49].

#### 3.1.2. SEOV in the Far East

##### SEOV in Japan

Of interest, all HFRS cases hitherto reported from Japan seem to have been exclusively rat-transmitted cases. From 1960 on, epidemics of a then unknown feverish disease, called “epidemic hemorrhagic fever” (EHF), were seen principally in the back alleys of Osaka and were characterized by “marked but transient proteinuria”, a salient feature of all HFRS presentations [1,50]. In 32 EHF cases, proteinuria peaked on day 6 post onset of symptoms (POS = sudden fever), and disappeared almost completely by day 12 in severe, and by day 7 in mild AKI cases [50]. The severity of proteinuria appeared to be a predictor for EHF clinical severity, a finding confirmed more than 50 years later in 70 cases of European PUUV. In this study, often massive proteinuria (1/3 in the nephrotic range) peaked by day 5 POS and disappeared almost completely by day 11, whereas serum creatinine values peaked on day 9 [51].

In May 1977, a link between EHF and Korean KHF, which had been rampant during the Korean War, was established by demonstrating in a blinded test HTNV-like antibody in the sera of EHF cases from Osaka, 7 to 17 years after acute EHF, but no antibody in the sera of healthy Japanese controls [52]. At least 120 residents in Osaka City were shown to have suffered from urban rat-transmitted EHF, including two with a fatal outcome in the 1960s, followed by five laboratory rat-transmitted EHF cases in the 1980s [53]. In a retrospective study, it appeared that already during the 1960s, 15/89 (16.8%) brown rats captured in the Tokyo port area and 6/59 (10.2%) brown rats in a Tokyo suburb were seropositive for this “HTNV-like” agent [54]. Two SEOV strains were isolated from these Tokyo urban brown rats and called JTRN/82/14 and TR-352 VE8 [26]. 

Rat-transmitted EHF outbreaks in Japanese research laboratories were an extensive and recurrent problem. In some animal rooms, e.g., at the Tohoku University in Sendai, 90% of 135 laboratory rats, and in Wakayama Medical College, Wakayama city, 88% of 117 laboratory rats were reportedly infected by a “HTNV-like” virus ([55], pp. 820–821). This infection was also reported in apparently healthy purchased brown rats, although infection was denied by the rat-breeding companies, since rats appeared to be in persistent good health. In the 1970s, at Tohoku University Hospital, Sendai, 14 scientists developed serologically confirmed HFRS that was clinically mild, except in one case. Since this institution had witnessed several recurrent outbreaks previously, the mysterious affection, originally called “Sendai hemorrhagic fever” was re-named KHF, as patients had detectable “KHF-like antibody”. The antibody was also present in three animal technicians who lacked a clinical HFRS-history, but who had worked in the same room that housed the infected rats [56]. The clinical picture in infected humans differed, however, from classical HFRS by a more pronounced hepatic, rather than a renal involvement, except for a massive initial proteinuria, often over 3 gr/24 h [53].

A prototype Japanese SEOV strain was isolated in 1983 from laboratory rats, called “Sapporo rat” or SR-11 [21]. By the end of 1985, Japan had registered 129 laboratory-acquired HFRS cases from at least 17 animal rooms, which caused great concern in the Japanese press and TV [57]. In 1982, at the WHO Collaborating Centre for Hantaviruses in Seoul, Korea, most of these rat and human samples were confirmed as SEOV-infected, and HW Lee’s laboratory was officially designated to monitor laboratory animal colonies of breeders in Japan ([55], pp. 826–827). Laboratory researchers reportedly contracted HFRS more frequently than animal technicians or caretakers, although an animal caretaker died [57]. Another SEOV strain, called B-1, was isolated in 1983 from a laboratory rat tumor in an institution where a HFRS case had occurred [58], in striking similarity with the later SEOV findings in the UK (see below). On an island in Tokyo Bay, 99/355 (27.9%) of wild brown rats were found to be IFA-positive in screening with the local Japanese SR-11, and a SEOV isolation called TR-352 was made [59]. Another SEOV strain, KI-83-262, was isolated in 1983 from a wild brown rat captured on a dumping ground in Hokkaido [60]. Rats were sampled from 1983–1992 in the same Tokyo Bay location and further PRNT-confirmed SEOV isolations were made, including TQR-23, TQR-48 and TQR-50. This study showed the multi-year persistence of SEOV infection in local urban brown rats in Tokyo, in contrast to an originally putative recent local introduction [61]. This surprising finding was later confirmed in several other follow-up studies in different countries, including the USA (see below). 

Heavy proteinuria and hepatic or coagulation perturbations, rather than notable AKI, were observed in 11 cases occurring in Nagoya City University Medical School, stressing again the peculiar clinical presentation of this “HFRS” form [62]. As a further proof that SEOV infection was not only present in laboratory rats, but also in wild commensal brown rats of the same Nagoya City, a SEOV strain was isolated from a brown rat, captured in 1990 in a residential area and named NR-9 [63]. In 1994, comparison by RT-PCR of M genome nucleotide sequences found a homology of 93.4 to 98.1% in SEOV strains derived from laboratory rats (SR-11 and B-1), as well as from urban brown rats captured in Japan (KI-83-262), Korea (Seoul 80-39) and China (R22, see below) [64]. This genetic relative homology was confirmed later in other SEOV strains despite geographic and even worldwide dispersion, thus hampering the determination of the exact geographic area of introduction or of human SEOV infection, a challenge that persists even today [22]. 

In summary, the peculiar clinical presentation of a milder HFRS form, with marked but transient proteinuria and heavy hepatic involvement, with wild and laboratory sources of infection, and with diagnostic serological tools (IFA, later supplemented with PRNT) was described in Japanese literature almost four decades ago. At that time however, disease names like KHF in Korea or the EHF in China were used to designate SEOV-HFRS, even though these epidemic affections should more accurately be restricted to illness caused by the Korean prototype HTNV [20]. Taking into account the magnitude of the (SEOV-)HFRS problem in Japan, the first international and WHO-sponsored meeting on HFRS took place in 1982, in Tokyo, Japan [14]. Although PRNT-confirmed SEOV infections were found in some Japanese cases with hepatitis of unknown etiology [65], and despite Japan isolating several distinctive SEOV strains from different rat sources, not a single new HFRS case has been reported from Japan since 1984. This was in spite of SEOV-infected wild brown rats being continuously found throughout Japan [66,67]. This situation was virtually the same worldwide for over three decades, until the recent rediscovery of SEOV-HFRS in the West since 2012 [13]. This observation underlines the diagnostic difficulties in detecting mild and “atypical” HFRS clinical disease in the Far East and probably worldwide, including on the African continent [1,7,68].

##### SEOV in China

China was and still is by far the country most affected by HFRS epidemics, formerly called EHF. EHF epidemics were recorded in China since the 1930s. During a noticeably long registry period, 1950–2014, the National Health and Family Planning Commission recorded a total of 1,625,002 EHF cases with 46,968 deaths, or an overall case fatality rate of 2.89% [69]. The etiological relationship between Korean KHF and Chinese EHF was first shown in 1980 [70], both supposedly caused by the Korean prototype HTNV. However, the co-occurrence of the milder “*Rattus*-type HFRS” was also identified serologically in the early 1980s after the isolation of two Chinese SEOV strains, called R22 and K24, both from wild brown rats captured in the houses of Chinese EHF patients with frequent rat exposure from the rural Henan province [20]. Rat-induced epidemics explained a predominant but different “mild type of HFRS” in several regions in China, as was recognized already decades earlier on clinical and epidemiological grounds alone [71]. Moreover, higher IFA titers from the same mild HFRS cases were obtained in sera tested with the lung antigen from these “house rats”, versus lower IFA titers obtained with the Korean prototype HTNV [71,72]. In contrast to severe HFRS induced by the prototype HTNV, these SEOV-HFRS cases showed less hemorrhages, but increased liver involvement [71,73]. Moreover, in the early 1980s, Song Gan et al. already remarked on an often milder AKI, with a shorter clinical self-remitting course of approximately two weeks [72] and the lower fatality rate of approximately 1% in SEOV-HFRS [69,72]. These observations were similar to those made in Japan [50,53] and Korea [28], and heralded most of the comparable later findings in the USA [2,3,4], North Vietnam [43], and France [7]. In fact, the same generally good prognosis, and rapid self-remittance within two weeks without sequels, was later confirmed in most of the equally mild, but PUUV-induced, European HFRS cases [51,68,74]. SEOV infections were considered in some Chinese provinces to be more frequent than HTNV infections [75], especially in residential areas. Indeed, clinically inapparent SEOV infection rates in “Rattus-type HFRS” regions could involve 8% up to 20% of the local population—an interesting 1999 finding probably still applicable today to the current western pet rat SEOV problem [73]. Finally, several HFRS outbreaks, mostly mild, but always involving people working with infected laboratory rats, were reported from different Chinese laboratories from 1985 on [76,77,78]. Up to 2000, China counted at least nine different SEOV isolations from *R. norvegicus* (strain L99, however, being from *R. losea*), and six SEOV isolations from humans [79] (See Figure 1, Asia). This dual HTNV and SEOV infection pattern prompted mass vaccination campaigns in China, using a combined inactivated vaccine that has now been used with success for decades [75,80]. In Taiwan and in the early 1980s, 31/240 (13%) of humans were found to be IFA HTNV-positive, of which were two HFRS cases [39]. Hitherto, SEOV remains the only confirmed hantaviral pathogen reported from Taiwan.

Since milder SEOV cases can be easily missed clinically and may thus represent diagnostic challenges [1,68], some authors suggested that SEOV might in fact be the most common hantaviral pathogen in China [33]. As China accounts for over 90% of cases noted globally [75], SEOV might consequently be the most important, but paradoxically also the most underestimated, hantaviral pathogen worldwide.

#### 3.1.3. SEOV in South-Asia

In Sri Lanka, 13/96 (13.5%) brown rats captured in 1987 in Colombo harbor was SEOV antibody-positive by PRNT and a novel SEOV strain, called Sri Lanka R-1315, was isolated ([55], pp. 1747–1748) and [81].

In 248 patients with leptospirosis-like illness in 1987, 14 (5.6%) appeared “HTNV antibody-positive”, however, with equal PRNT titers for SEOV and for HTNV, suggesting a possible third murid pathogen (THAIV?). Four cases were also ELISA IgM-positive and all had mild renal involvement, one with lung symptoms, another with prominent hepatitis, and a third fatal case with meningo-encephalitis [81]. Of note, 8/105 (7.6%) other leptospirosis-suspected patients from Kandy, Sri Lanka, were likewise ELISA HTNV-seropositive, but showed clearly higher titers in PRNT for THAIV than for SEOV, implying a pathogenic role for THAIV [82].

In May 1985 [39], 1/98 (1.1%) of human sera and a HFRS patient from India were “HTNV antibody-positive”, whereas 15/204 (7.3%) of unknown hemorrhagic fever patients from the Indian Andaman and Nicobar islands in 1990 reacted positive to the SEOV Sri Lanka strain R-1315, together with 11/90 (12.2%) local wild brown rats ([55], pp. 820–821). A prospective 2006 study of South-Indian patients, hospitalized in Cochin and Chennai with an acute leptospirosis-like illness with fever, AKI, “transaminitis”, and thrombocytopenia, found they were negative by micro-agglutination test and Patoc ELISA for leptospirosis, and ELISA-negative for dengue fever. However, 7/60 (12%) patients were positive in both the SEOV stripe immunoassay (SIA) and SEOV IgG ELISA. Six of these were also positive in IgM SIA, confirming a recent infection. Of note, 3/60 (5%) of these other Indian cases in contrast showed a clear PUUV-like SIA pattern, suggesting a cross-reaction with an as yet unknown (arvicolid?) hantaviral pathogen, putatively present in the Indian subcontinent. Moreover, 2/3 of these PUUV-like cases were fatal, despite combined acute hemodialysis and ventilation [83]. RT-PCR performed later on all seropositive, but often thawed, samples found no hantaviral viremia [84].

In the Philippines, 51.5% of 167 Manila seaport rats, and 5% of 400 human sera were found IFA HTNV-seropositive in 1983 [39]. In a later study of 461 asymptomatic study subjects, using a high-density particle agglutination (HDPA) technique based on a HTNV 84/105 screening antigen, the seroprevalence was about the same in males (6.1%), females (6.1%), and rural (7.6%), urban (5.6%), or urban poor (5.1%) populations, suggesting possible domestic rat exposure [85]. A first report of SEOV-HFRS and RT-PCR proven SEOV infection in urban rats followed from Bandung, Indonesia [86]. In a subsequent study in Indonesian patients with FUO, two cases of serologically and RT-PCR proven SEOV infection were described, with severe “transaminitis”, however, without any evidence of concomitant AKI [87].

### 3.2. SEOV in Oceania

In Australia, high antibody prevalence rates using HTNV-IFA screening were found in wild brown rats captured between 1981 and 1983, with 2/16 (13%) positive in the Northern Territory, 2/12 (17%) in South Australia, 3/8 (38%) in Victoria, and 8/19 (42%) in Queensland [11]. However, no typical HFRS cases were reported from Australia through 2005 despite documented HTNV-seropositive rodents [88]. HTNV-like antibody-positive human and rat sera were found in Fiji between 1981 and 1985 [39]. A serological survey performed by IFA among 61 wild brown rats and 199 patients from French Polynesia confirmed this evidence of “HTNV-like” infection in one rat and in two patients [89]. We found no data concerning New Zealand. In Hawaii, 8.8% of 1482 urban brown rats and 6.9% of 252 human sera appeared IFA HTNV antibody-positive [39].

While the various serological assays used differ from study to study and lacked specificity, it is clear that these early observations from Asia and the Pacific regions suggest that hantaviruses, most probably SEOV, were present in rodent populations, and likely caused human infections. 

### 3.3. SEOV in Europe (Figure 2)

Although wild rats, serologically proven since the 1980s to be infected with a “HTNV-like” patholological hantaviral agent in many European countries [10], scientific interest in their potential impact on public health resurged only recently, reconstituting SEOV-induced HFRS as an “emerging infection”(Figure 2).

#### 3.3.1. Russia

A large laboratory outbreak of HRFS-like disease in 1962 affected 83 technicians in Moscow and led to suspicion that this then unknown febrile disease may have been spread by aerosols from infected laboratory rats [96]. 

#### 3.3.2. SEOV in Belgium and the Netherlands

In the late 1970s, three laboratory outbreaks of a new renal disease with fever, occasionally necessitating acute dialysis, were all associated with handling and processing asymptomatically infected laboratory rats. First described in a Belgian laboratory [92], they occurred almost simultaneously in a Dutch laboratory involving four HFRS cases [97], and in a British laboratory [90,98]. Albino brown rats, called Lou (from “Louvain”)/M/Wsl rats, and rat immunocytomas, were supplied upon request from the Unit of Experimental Immunology, Catholic University of Louvain, Brussels, at a time when hantavirus infection of laboratory rats was unknown and unsuspected in Europe. Once diagnosed, these became the first cases of human illness due to a rat-borne “HTNV-like” virus and of hantaviral infection in laboratory rats, documented outside East Asia [92]. Infectious aerosols may have been generated from infected animals or tissue processing, resulting in the human infections within the three affected laboratories. The origin of the virus was not identified, but infected rats were removed. Inspection of the three other Belgian institutions found no signs of infection among either staff or rats. The specific diagnosis of SEOV infection was technically still impossible in 1983, but the pathogen was found to be distinct from PUUV [92,99]. The virus was thought to have been introduced between 1970 and 1974, and infected half of the 60 technicians working in the Brussels laboratory. Similar HTNV-like IFA antibodies were retrospectively demonstrated in rats’ serum, inbred during 1973–1982 [99]. Most of the infected technicians did not recall a HFRS-like illness; however, three required hospitalization in 1978. The most severe case (peak serum creatinine 8.8 mg/dL) needed three dialysis sessions and prompted a kidney biopsy. The patient presented very mild (0.2 g/L) proteinuria on admission on day 3 POS, which peaked with the marked nephrotic range (19 g/L) on day 6 POS, and disappeared completely by day 19 POS [93]. This characteristic fugacity in proteinuria is typical for SEOV-HFRS [50], and probably for all HFRS forms as well [1,19,68]. It also indicated, 40 years ago, the necessity of repeated, or even daily, urine examinations for assessment of the full magnitude of this frequent (but sometimes sole) form of renal involvement in HFRS [51]. Transient microhematuria was also documented. The remarkably similar symptomatology of these three AKI cases, all rapidly transient, led Belgian nephrologists to report the first (but still surprisingly valid) Western purely clinical description of SEOV-nephropathy in 1979 [93], i.e., three years before the publication of the causative pathogen itself [8]. This new scenario of HFRS originating from exposure to infected laboratory rats was recently called a “second twist of SEOV exposure” involving unusual interactions with rats, outside the realm of exposure to wild *R. norvegicus,* with a “third twist” being the exposure to SEOV-infected pet rats [13]. 

A serological report of European human SEOV infection transmitted by laboratory rats followed in 1991, involving 11 asymptomatic Dutch and Belgian laboratory technicians working with rats in the 1980s [100]. In contrast, 14 non-laboratory but symptomatic “wild” HFRS cases, likewise in the 1980s, appeared to be PUUV-infected. Interestingly, two “wild” acute HFRS cases were only reactive to Japanese SEOV strain SR-11, and not to PUUV, as confirmed in a then novel competition (blocking) ELISA test. Consequently, these two cases, a Dutch farmer and a Belgian homeless person, both with exposure to wild rats, were among the earliest serological demonstrations in the West of wild rat-borne SEOV-HFRS [100]. Of note, of a total of 4750 rodents captured between 1986 and 1990 in Belgium, the Netherlands and Germany, the brown rat was the second most IFA HTNV-seropositive species, (2.1%), after the bank vole with 14.4% seropositivity [10]. Among 502 brown rats trapped from 2004 to 2006 in the Flanders region (Belgium), IgG against SEOV was found in 129 (27.1%). A molecular investigation performed on six seropositive rats allowed the detection of one SEOV strain [101]. Of 16 wild brown rats trapped in 2013 in the Eastern province of Gerderland, three were found to be infected by SEOV using serological (PRNT) and molecular assays [102]. Moreover, the first proven Dutch case of SEOV-HFRS was recently documented [103].

#### 3.3.3. SEOV in the UK

Between January and July 1977, four technicians of the Institute of Cancer Research (ICR), Sutton, UK, became acutely ill with typical HFRS symptoms, some with acute lung injury. Three cases required hospitalization and two needed acute dialysis due to anuria [98]. The Catholic University of Louvain in Brussels that had a SEOV laboratory outbreak one year later (see above) had sent apparently healthy but in fact SEOV-infected LOU rats, and their implanted immunocytomas to ICR and other laboratories in Oxford, Sutton, and Bristol between 1974 and 1977. Other early reports of similar laboratory outbreaks in Korea, Japan, and particularly in Belgium, signaled a breakthrough, and the immunocytomas, which had been stored for eight to ten years at −70 °C in the Special Pathogens Reference Laboratory, Porton Down, Wiltshire, were thawed. This led to the isolation of five SEOV viruses in April 1984, cultured on Vero E6 cells, and labeled H95 to H99. They were later renamed after the Belgian immunocytoma classification, bearing the French abbreviation “IR” (for “immunocytome de rat”), as IR22 to IR1060 [90]. Some of these novel viruses were again renamed “GB-B” (for “Great Britain”) and were used in IFA for diagnostic purposes [99,104,105]. A retrospective 1984-5 investigation of ICR personnel revealed that 9/15 (60%) technicians working with Belgian rats and/or rat immunocytomas had markedly higher IFA antibody levels against HTNV and the novel UK hantavirus strains H96, H97, H98, and H99, as compared to a recently isolated Russian PUUV strain, CG 18-20. SEOV-HFRS incurred by working on infected immunocytomas represented thus another form of “second twist of SEOV exposure”, i.e., exposure with no direct contact with rats [13]. Belgian sera of rats with these implanted immunocytomas showed the same HTNV-like reaction pattern, leading to the retrospective conclusion that the British 1977 HFRS outbreak was likely due to a laboratory rat-transmitted hantavirus [90]. This was confirmed using monoclonal antibodies that demonstrated that GB-B strains belonged to the “Rattus-type” hantaviruses [104]. A total of 19 years later, isolate IR461 derived from an immunocytoma imported from Belgium to the Sutton ICR was finally sequenced [106]. Since the ancestors of the Belgian LOU/M/Wsl rats had been housed in a Japanese animal facility in Hokkaido, where SR-11 had been isolated, a close relationship between IR461 and this Japanese laboratory rat strain was anticipated. However, a close and unexpected clustering with two wild rat American SEOV strains, Baltimore rat virus and Belem (Brazil), was found. These formed a unique clade grouping laboratory strain SR-11 with two wild rat American SEOV strains, Girard Point virus and Tchoupitoulas (see below) [106]. The 1984 discovery of a SEOV strain in the UK was the earliest isolation of a murid hantaviral pathogen in Europe and occurred at a time when similar pioneer findings were being made in the Americas and in Africa, all during the early 1980s (see below).

A clinical series of 15 HFRS cases, mostly farmers hospitalized in the UK from 1989–1992 with fever, AKI and thrombocytopenia (two with icterus), plus one asymptomatic control subject (also a farmer) constituted the second piece of evidence in Europe of SEOV likely spread to humans by wild rats, this time in Northern Ireland. Since Northern Ireland and Portugal are the sole regions in Europe with the wild rat as the only known rodent reservoir of hantaviral pathogens, the almost exclusive IFA and ELISA reactions to the Chinese (Henan province) wild rat R22 strain were deemed evidence of acute SEOV infection [94]. This was later confirmed when rodents captured around patients’ farms in County Down, Northern Ireland, yielded a high seroprevalence in 11/51 (21.6%) of wild brown rats, when tested with the same SEOV R22 by IFA. This constituted a first warning of a new potential public health problem caused by SEOV in the UK. [107]. Of interest, a Yorkshire famer was later proven by IFA and SIA to have contracted SEOV-HFRS. Like his Northern Irish colleagues 20 years earlier, leptospirosis was originally suspected, but not confirmed [108]. However, a brown rat on his farm yielded in RT-PCR an amplicon having 97% nucleotide sequence similarity with the British laboratory rat SEOV strain IR461, and resulted in a second British SEOV isolation, called Humber virus. For both procedures, the same Chinese R22 strain, as had been used in 1993 and in Northern Ireland, was used as the positive control [5]. 

Since 1983, a number of serologically confirmed HFRS cases have been reported in the UK with probable or known associations to wild or pet rats, reviewed in McElhinney et al. A study of epidemiologically related infections in pet rats and pet rat owners suggests a high prevalence of SEOV within UK pet rat populations [95]. More recently, a survey of wild rats on farms in Northern England has detected SEOV RNA in 19% (13/68, 95% CI 11 to 30) of rats and 12/13 sequences had 98% nucleotide identity to the GB Humber strain of SEOV. Unlike in France, all UK SEOV sequences (laboratory, pet and wild rat SEOV) to date have been exclusively associated with SEOV lineage 9 [109].

#### 3.3.4. SEOV in Germany

The first SEOV isolation on the European continent was performed by J. Pilaski et al. from a wild brown rat, caught in 1988 in Polle, Lower Saxony, Germany, and was called “Polle 18” [91]. We found no record of this German hantaviral isolate being sequenced, but it was confirmed as being a novel SEOV strain in the WHO Collaborating Centre for HFRS in Seoul, where in 1986, another SEOV-positive rat from Germany was documented ([55], p.581). Recently, a SEOV infection was diagnosed in a 70-year-old German tourist returning from Sulawesi, Indonesia [22]. The initiating fever was situated before his departure from Indonesia, over 3 weeks before his eventual hospitalization for HFRS with AKI back home in Hamburg, Germany. However, the rapid evolution of his laboratory parameters, with, e.g., still rising blood levels of creatinine and leukocytes on day 2 and 3 of hospitalization, suggested that he might have been infected more recently in Germany, after his return from Indonesia, particularly in view of his still positive SEOV viremia, i.e., over 3 weeks after the presumed onset of fever.

#### 3.3.5. SEOV in Portugal, Greece, France and Sweden

The first (1993) HFRS case published in Portugal was a combination of mild non-oliguric AKI (peak serum creatinine 4.73 mg/dL), but with heavy hepatic involvement, including pronounced icterus (total bilirubin 23.42 mg/dL), both showing a spontaneous total recovery. This acute case was negative in leptospiral serology but was serologically positive for SEOV [16,110]. SEOV remains the only known hantaviral pathogen in that country [111]. The first IFA-SEOV-positive wild rat in Greece was documented in 2000, albeit at a low titer (1/32) and with negative RT-PCR [112].

In France, in the 1980s, 19 of 70 (27%) Albino brown rats LOU/M/Wsl were found by IFA to carry IgG against a HTNV-like virus. They were also bred in a Paris animal house for more than ten years. Three animal caretakers were found seropositive, but none reported any HFRS disease [113]. In 1985, a HFRS case with icterus from the South West of France, exposed to wild rats, showed an IFA seroconversion against HTNV, while the result was negative against PUUV, suggesting retrospectively a SEOV infection [114]. A total of six HFRS cases, dating back to 1977–1996 was reported as serologically suspected for a SEOV-infection [7,115] and one more in 2014 [116]. The first (2012) RT-PCR proven SEOV-HFRS case in Europe was found in France in a pregnant woman, with hepatic and renal complications that were clinically almost undistinguishable from pregnancy-induced autoimmune liver disorders, complicated likewise by thrombocytopenia [117,118]. From 2014 to 2016, three additional SEOV-HFRS cases followed in France, again with more hepatic than renal involvement. One case had perfectly normal renal function parameters; however, clear thrombocytopenia was initially present in all four proven SEOV infections [7].

The French SEOV-FHRS cases were associated with wild brown rats, pet brown rats and reptile food brown rats, in which SEOV was also detected [7]. Previously, RT-PCR evidence indicated the presence of SEOV in French reared brown rats, sampled in 2003, and in the wild from Lyon and Orleans regions [119]. The presence of SEOV in wild brown rats from Lyon region was confirmed later when 18/128 animals (14%) were found positive by RT-PCR [120]. Finally, a first French SEOV isolation was recently obtained from a wild rat captured in Mantenay Montlin, Central France (Figure 2, Europe) [121]. There is also the RT-PCR detection of SEOV in a pet rat from Sweden in 2013. The animal was tested on the request of the owner. It was imported from the UK, but the origin of the SEOV remains unclear [122].

Except for the more recently discovered Tula orthohantavirus (TULV*),* these results document the early detection of at least two pathogenic hantaviruses in European rodents, likely to be SEOV and PUUV, and demonstrate associated human infections from both laboratory and pet rats, and from rats existing in the wild. The above findings suggest that these viruses are enzootic throughout many parts of Europe, and remain a continuing but hitherto underestimated threat to human health. 

### 3.4. SEOV in the Americas (Figure 3)

Human SEOV-induced HFRS cases were first and most reported from the Baltimore, MD, region (See text and Table 1), although urban and wharf rats, proven in PRNT to be infected with SEOV, were often demonstrated throughout the USA, and later likewise in South-America, wherever tested from the early 1980s on (Figure 3).

#### 3.4.1. SEOV in North America

In the USA, 1982 was significant with the discovery of SEOV in the New World. A “HTNV-like agent”, different from the European PUUV, was described in wharf rats in New Orleans, Philadelphia and Houston [124,125]. In Baltimore, Maryland, up to 74% (32/43) of brown rats showed the HTNV-like IFA reaction, as did three renal patients [126]. With these simultaneous observations, it was suggested that a novel viral pathogen with renal tropism might have been introduced locally by international shipping, a hypothesis later repeated in several other countries where HTNV-like seropositive brown rats were discovered in port cities. However, rat infections were documented throughout inland USA as well, a finding not supporting a recent introduction. As confirmed by the Center of Disease Control (CDC), infected brown rats seemed to be omnipresent in the USA: *“Virtually everywhere we looked-West coast, East coast, Cincinnati, New York City–we found (SEOV) infected rats*” [127]. Of interest to the recent American pet rat observations, several serosurveys in 1985 reported no evidence of HTNV-like infection in brown rats in large-scale breeding colonies [128] or in cell-lines of brown rat origin [129]. However, in 1985, a research technician working for five years in Frederick, Maryland, on laboratory rodents (including brown rats) with virus-induced tumors, showed stable antibody titers to prototype HTNV by PRNT but not against PUUV. This 31-year-old blood donor had no relevant travel history, and did not recall any prior acute illness, suggesting that his infection was locally acquired and asymptomatic [130].

Successful isolations of SEOV strains followed from 1984 on, all from wharf rats, respectively captured in Philadelphia and Houston, called Girard Point virus (GPV) [9], in 1985 and in New Orleans (Tchoupitoulas virus or TCHV) [128], and in 1987, in Baltimore (Baltimore rat virus, or BRV) [131]. GPV was found by PRNT as being clearly distinct from HTNV [9]. In 1988, prior SEOV infection with local BRV was documented by PRNT in four life-long residents of Baltimore without a relevant travel history; however, no clinical evidence of a typical HFRS episode was recorded, again suggesting such rat-induced infections could occur asymptomatically [132]. Finally, on all continents, including Africa, Australia, and Europe (Malmö, Sweden), infection was demonstrated in 243/910 (26.7%) of peridomestic rats, confirmed by PRNT to be related to GPV, but not to HTNV [11]. Baltimore City, however, remained the focus of American hantavirus research, with at least three ensuing studies confirming the seroprevalence of approximately 50%, and up to 57.7%, SEOV-infected brown rats, and a residential brown rat population density remaining apparently unchanged during estimations in 1949, 1952 and 2004 [13,132,133,134,135,136]. A survey of the residents of Baltimore found that nearly 2/3 of respondents (64%) observed rats in streets and alleys, 6% saw rats inside residences, and 1.2% had experienced a rodent bite in their lifetime [133]. Some of the Baltimore SEOV-seropositive study subjects admitted that part of their daily home maintenance was to sweep rat feces from the paved areas of their properties [13]. Nevertheless, the striking absence of noticeable HFRS in the USA, in contrast to China, was recorded [13,19,137]. Following the extensive 1982 serosurvey of wharf rats in seven American port cities [124], another novel SEOV was characterized in a New York City wharf rat, and called NYC Baxter virus [138]. This virus was announced as probably recently introduced into New York City, based on a “time to most recent common ancestor” (TMRCA) calculation, which included the recently isolated (2012) British wild Humber strain [5]. However, an argument against a recent SEOV introduction is that SEOV-infected brown rats were documented in New York City (NYC) in 1985 [127,128], whereas predictions of the degree of rat infestation in several NYC boroughs, based on the number of recorded rodent bites, did not show substantial changes between 1986 and 1994 [139]. In a recent overview, wild rats in the USA were described as being SEOV-positive in 13 states, from Alaska to Mississippi and Hawaii. SEOV endemicity was confirmed by RT-PCR in New Orleans wharf rats, apparently unabated over 30 years, i.e., since the 1985 isolation of TCHV virus from the same city [140]. In Canada, 29/2063 (1.4%) human sera were found to be IFA HTNV-positive in the early 1980s [39], suggesting SEOV infection from wild (or domesticated?) rats, long before the local 2017 problem of SEOV-infected pet rats became (re)apparent.

#### 3.4.2. SEOV in South America

GPV-positive brown rats were also found in Argentina and Brazil (6% in Recife and 65% in Belem), resulting in the isolation of the first South-American hantaviral pathogen, Belem virus [141]. Two SEOV PRNT-positive human sera were found in 1994 and in Brazil, one of which with a high antibody titer in a resident of Sao Paulo, who had never travelled abroad, and with a presumptive diagnosis of leptospirosis [142]. Four of 201 Brazilian leptospirosis-suspected samples from Belem were found in 1998 concomitantly IgM-positive for HTNV and for leptospirosis, suggesting a recent dual infection with two different pathogens spread by wild rats. Two of these four sera were examined by PRNT with one showing a low, and the other a strongly positive titer (1/320) for SEOV [143].

Salvador is a port city in North-East Brazil, surrounded by rat-infested slum dwellings with a very high leptospirosis endemicity. The continued presence of leptospira and SEOV was demonstrated in commensal rats trapped during the rainy season of 2010, and complemented with unpublished leptospira and SEOV prevalence information collected in 1998. SEOV was detected in 18% of wild brown rats in both 1998 and 2010 (*n* = 78 in 1998; *n* = 73 in 2010), indicating that brown rats in Salvador serve as permanent reservoir hosts for SEOV [144]. Previously, 13.2% of 379 asymptomatic Salvador schoolchildren were found IFA-positive for a HTNV-like agent, but none for SNV, suggesting potential rat exposure from a young age [18]. During a 1999 outbreak of urban leptospirosis after floods in Salvador, a total of 326 AKI cases were detected, of which 39% were serologically confirmed, and 19% considered as probable leptospirosis cases [145]; however, the question remained as to which other reno-tropic infectious pathogen could explain the remaining 42% unconfirmed cases [17]. Dual concomitant human leptospirosis-hantavirus infections have been described in tropical regions endemic for leptospirosis [146], as well as in Northern European non-endemic regions [147]. The earliest such dual infection with AKI, in a non-tropical country, was reported from Glasgow, Scotland, in 1988 by G. Kudesia et al. [95]. Despite these previous findings, no HFRS cases have been reported from Salvador to our knowledge.

After the 1990 floods in the coastal Brazilian city of Recife, 8/156 (5.1%) hospitalized AKI cases with thrombocytopenia were originally considered as leptospirosis, but were IFA-positive with R22, by HDPA based on HTNV 84/105, and were IgM-positive by ELISA, suggesting recent SEOV infections. These patients were negative for leptospira and PUUV serology. These eight Brazilian SEOV-HFRS patients can thus be considered as being among the very first clinically and serologically proven cases of human hantaviral illness in the New World [123]. 

In 1990, in Buenos Aires City, Argentina, 22.5% of 102 laboratory albino rats in three different animal rooms were found HTNV-like–positive by IFA and PRNT, as were three technicians working with rats in the same animal rooms [148]. In the same city, 18 years later, SEOV seroprevalence was found in 11.9% of wild brown rats (*n* = 151), varying between 0 and 26.1% depending on the site. The authors concluded that rat SEOV infection was geographically widespread in Buenos Aires City, and had been enzootic there for at least 20 years. Nevertheless, no HFRS case had been noted from the city during that period [149]. 

#### 3.4.3. Conclusions For, and Table of American SEOV-HFRS Cases

Until recently, most reviews indicated that HFRS was relevant only for the Old World, but absent from the New World. However, in addition to the eight Brazilian SEOV cases, two fatal SEOV cases, one with lung involvement, were confirmed in Peru [150] and six clinically documented and laboratory-confirmed SEOV cases, infected by wild rats, were found in the USA, in an observation period stretching from 1994 to 2012 (Table 1) [12,151,152,153]. These USA SEOV-HFRS cases included one lethal case (case 6 [153]), with both renal and fatal pulmonary involvement, confirming the ability of SEOV to cause sometimes clinically severe, and even fatal, human infections in the New World. Moreover, one out of these six American wild rat SEOV-HFRS cases also showed signs of secondary hemophagocytic lymphohistiocytosis (HLH), a rare but often serious complication of mostly viral infections, as noted before in Old World HFRS cases [154,155]. 

### 3.5. SEOV in Africa (Figure 4)

Urban rats collected between 1981 and 1983 in Port Said, Alexandria, Egypt, and Mombasa, Kenya were IFA-positive for HTNV, and two rats from Egypt were confirmed PRNT-positive for GPV [11]. In 1982, 27% of 859 sera of urban rats captured in Cairo, Egypt, were found IFA-positive for HTNV, but negative for PUUV, including 42.2% of rats captured in Cairo Governate ([55], pp. 297–298). The first pathogenic hantavirus isolated from a local wild rodent in Africa was SEOV. Two different SEOV strains from wharf rats captured in Cairo were isolated in 1983–1984 and called Egypt R/12915, and Egypt R/13120 respectively, a finding in line with the demonstration of 19.9% (499/2,499) of wharf rats being positive for HTNV-like antibodies, as were 1.3% (6/458) of an asymptomatic human cohort, both from Cairo [39]. These data were later supplemented by H. Hoogstraal with 0.6% HTNV-like prevalence in 154 human samples, and 23% prevalence in 1846 wild rat samples, both from Egypt [156]. A similar presence of HTNV-like antibodies was demonstrated in rat and human populations of Djibouti [157]. Finally, the presence of a SEOV-like agent was found in a 2004 Egyptian study measuring anti-hantavirus IgG, using ELISA with HTNV as screening antigen, in 350 patients with end-stage renal failure (ESRF) and a group of 695 healthy controls. A prevalence of 1.4% was found among the ESRF cases versus 1.0% in healthy controls, i.e., a statistically non-significant difference of *p* = 0.48. According to a questionnaire, all antibody-positive study subjects (ESRF cases and controls) had been exposed to wild rodents [158] (Figure 4).

In Uganda in 1983, 15/355 (4.3%) serum samples of FUO patients were found to be HTNV-like-positive, together with 3/64 (4.6%) feral rats [39] and ([55], pp. 1907–1908). In Sudan, 28/352 (7.9%) urban rats were equally found HTNV-like-positive [39]. In 1984, Gonzalez et al. reported humans IFA-positive antibodies to a HTNV-like agent in four countries of Central and West Africa (Central African Republic, Burkina Faso, Benin, and Gabon). Four of the highest IFA titers for HTNV were, however, negative for the Korean prototype HTNV by PRNT, suggesting possible infection by another closely related (murid?) African hantavirus [159]. Between 1985 and 1987, sera from 5070 randomly selected persons living in six central African countries (Cameroon, Central African Republic, Chad, Congo, Equatorial Guinea and Gabon) were IFA-screened for antibodies to six different viral hemorrhagic viruses (VHF). The highest sero-prevalence after Ebola VHF (12.4%) was for HTNV, with 6.15% [160]. Of note, this prevalence of 6.15% is noticeably higher than for most currently known nationwide seroprevalence studies, even in countries long known as chronically endemic for hantavirus infections. Finally, Gonzalez et al. reported the presence of IFA HTNV-like antibodies also in Rwanda (0.4% of 280), Swaziland (3.4% of 202), Senegal (6.1% of 341) and Mauritania (2.4% of 82) [156]. 

Like in North America, South America, Japan, the UK, Germany, and Sri Lanka, where SEOV was also the earliest isolated hantavirus, the rodent reservoir of prototype HTNV, *A. agrarius coreae* (the Korean striped field mouse) is equally absent in Africa. This suggesting that these early serological findings using HTNV-like antigens may likely be ascribed to cross-reactions with SEOV, or a very closely related murid hantavirus [156]. 

More recently, Sangassou virus (SANGV) has been isolated and functionally characterized in cell culture as a novel second murid hantavirus species in Africa [161], Specific neutralizing antibodies against SANGV virus, were demonstrated in two FUO patients and in two other samples from Guinea, West-Africa [161]. Finally, an Abidjan, Côte d’Ivoire, patient with AKI was found IgM-positive to Bowé virus, another newly described hantavirus carried by an insectivore (not a rodent), Doucet’s musk shrew, but remained negative by RT-PCR [162]. Serological studies detected human hantavirus antibodies in West Africa, and a seroprevalence of 1.0% in the human population of the South African Cape Region (*n* = 1442). However, no clear-cut human pathogenicity was found caused by SANGV, nor by other novel hantaviruses, characterized in shrews and bats [163]. Clearly our understanding of the hantaviruses in Africa remains incomplete; however, human HTNV-like infections, previously documented in 17 different African countries, may suggest that SEOV is a substantial cause of hitherto unrecognized African HFRS, or even HCPS cases, given the omnipresence of wild rats, particularly in urban settings.

## 4. Current Situation

In 2012, a Welsh pet agouti rat owner developed a severe MOF, with a combination of AKI, progressive lung and liver involvement, coagulation disorders and lactic acidosis, initially ascribed to overwhelming sepsis, needing 38 days of ventilatory support and 21 days of renal replacement therapy, but ending in full pulmonary and renal recovery. Ultimately, SEOV infection was confirmed by IFA, but SEOV viremia was not detected [6]. Rat-borne SEOV was traced back to a pet rat breeding site in Cherwell, England, where two owners were apparently also infected in 2011, one asymptomatically, the other requiring hospitalization for AKI with thrombocytopenia, secondary to a then unidentified viral illness [5]. These cases launched an investigation in the UK into the extent of SEOV infection among pet or domestically bred rats and their owners or breeders. IFA SEOV-IgG seropositivity in 27/79 (34.1%) was found among breeders versus much lower IgG prevalences in all other control groups (farmers and other professions with rat-exposure), encompassing 844 human serum samples. No clinical complications were found [164]. By contrast, the extended Cherwell rat study revealed SEOV RNA presence in different organs of 17/21 (81%) rats, and anti-SEOV antibodies in 100% (20/20) in blood tested by PRNT and ELISA [95]. These findings were much higher than the preliminary RNA RT-PCR screening of Cherwell rat blood alone, which found 7/21 (33%) positivity [5]. Partial L segment sequencing of SEOV viruses from Cherwell and Cheltenham (another English pet rat SEOV strain) clustered closely together with two wild rat strains, American NYC Baxter and British Humber, and with IR 461, a lab rat strain [95]. 

In France, the link between infecting rats and human HFRS cases was demonstrated by highly similar nucleotide sequences derived from both patient and rats [7]. After a SEOV-HFRS outbreak in 2006 among eight students of the Shenyang Pharmaceutical University, Liaoning Province, China, phylogenetic analysis revealed that the same SEOV strain had affected both patients, laboratory rats, and wild rats, the latter captured on the grounds of the concerned pharmaceutical building. This confirmed for the first time the triple scenario that SEOV-infected wild rats can infect laboratory rats, which in turn can infect humans [78].

This paucity of SEOV cases in the West and in Africa contrasts with the long-standing recognition of cases in the East, particularly in China, even since the early 1980s. The prevalence of SEOV-infected rats is similar in many parts of Asia to that found in Europe, the Americas and Africa, and appears to have been fundamentally unchanged over decades. 

Human SEOV-HFRS has until recently (2012) been underestimated in the West [1,16,17,18,19,68,165]. The reasons for this are not clear, but differences in the degree of exposure and the availability of diagnostic testing probably play a role: in China, virtually all HFRS cases are serologically screened with HTNV, which cross-reacts with SEOV. In contrast, in Europe, serologic screening relies primarily on PUUV antigens [7,19], while in the Americas, SNV and/or ANDV antigens are most frequently used, and HTNV and/or SEOV antigens are not [16,18,22,94], thus probably missing some isolated SEOV cases. In parts of Europe, Dobrava-Belgrade orthohantavirus (DOBV), another murid pathogen isolated in 1988, can cause important cross-reactions with SEOV in IFA and ELISA serology [22]. Serologic analysis using two commercial European immunoblots was misleading in the first reported SEOV-HFRS case in Germany. The commercial Hantavirus Profile 1 Immunoblot (Euroimmun, Lübeck, Germany) lacked a SEOV antigen, and the *recom*Line HantaPlus IgG and IgM blot assays (Mikrogen, Martinsried, Germany) showed reactivity for the three murid pathogens DOBV, HTNV, and SEOV, but yielded the weakest result for SEOV nucleocapsid protein. The final diagnosis was only possible after RT-PCR examination on the earliest available acute serum sample after hospitalization in Hamburg, Germany [22]. In this aspect, it is useful to remember that if the IgG IFA screening of the first British series of North-Irish farmers with acute hantavirus-induced AKI had been performed only with the then classical antigens PUUV and HTNV, but without the Chinese SEOV antigen R22, 13/15 (86.7%) of the cases would have been missed in 1994, which is unacceptable [94]. 

In contrast, Canadian and British military personnel, both originally suspected of severe SEOV-induced MOF during the Bosnian War, were later shown to be in fact DOBV infections [166]. Another possible explanation for the striking East–West difference in SEOV-HFRS incidence is that American SEOV strains might be less virulent than their Asian counterparts [133]; however, SEOV phylogenetic trees show American SEOV strains clustering closely with Asian strains. In a recent South-Korean study applying next generation sequencing, multiple strains of SEOV of worldwide origin were examined. A phylogenetic tree of sequences of small RNA segments revealed a grouping of strains from the UK and the USA, clustering together with another group containing American Tchoupitoulas, Japanese Sapporo and Korean strains, including prototype SEOV 80-39. These findings are not suggestive of a fundamental difference between infecting wild rat strains (e.g., Humber and NYC Baxter), lab rat strains (e.g., Sapporo and all the British IR strains), or pet rat strains (e.g., Cherwell). [167]. This may be to be expected, since all are carried by the same rat species, *R. norvegicus*. Nor was the phylogenetic tree suggestive of a recent introduction of all these rat strains in the respective countries, since IR 461 was isolated in 1984, and stemming from Belgian laboratory rats, that were presumably infected with the same SEOV already in the 1970s [95]. In summary, the paucity of recognized SEOV cases outside Asia may reflect insufficient medical awareness of the “atypical” and often very mild aspects in SEOV-HFRS cases, implying even HFRS cases with preserved normal kidney function throughout [1,87]. JE Childs, familiar with the wild rat problem in Baltimore, MD, describes the current situation as follows: “physician awareness of this rare disease is negligible, and public health surveillance non-existent. Acute illness caused by SEOV, does not have a unique presentation, and signs and symptoms are consistent with a wide range of diseases” [13]. 

The current problem of SEOV infection in pet rats likely arose from SEOV-infected wild brown rats, visiting commercial or domestic ratteries, rather than stemming from laboratory rat colonies [106], although the latter scenario was also proven by RT-PCR in China [78]. Laboratory rats today are generally free of SEOV infection due to strict adherence to the total eradication of infected rats [57,92], followed by a worldwide policy of continuous SEOV screening, imposed traceability of rat batches, and/or other preventive techniques [7,99]. These policies are probably not in place in small, private ratteries [7], and in view of the sometimes high SEOV prevalence in wild brown rats, it is not surprising that infected domesticated rat colonies became established, and human infections followed. The lesson learned from recent British and American pet rat investigations is that pet rat owners and breeders are at increased risk of SEOV infection due to frequent close contact with their companion animals, but when present, acute disease is rarely severe or life threatening. This is based on the repeated observation that SEOV infection usually resulted in very mild illness [3,103], without mention of renal involvement [2,164], but with the possible exception of missed initial proteinuria [1,50,53]. Since proteinuria and even all degrees of concomitant AKI are almost always spontaneously self-remitting within 2–3 weeks in most cases of HFRS [6,22,68,74,103,151,152], the initial fear that rat-borne SEOV represents a newly discovered public health problem might not be justified. A heightened clinical acumen for the mild and/or atypical presentation of SEOV-HFRS is, however, clearly needed in the West. 

## 5. Conclusions

Since the 1976 original discovery of HTNV, a large body of evidence has accumulated over the ensuing four decades regarding the global distribution and clinical characteristics of hantaviral infections in both humans and small mammal hosts.

With increasingly sophisticated technology and enhanced clinical awareness, many specific hantaviruses are now recognized; however, SEOV remains the most widely distributed hantavirus. SEOV is maintained in nature among chronically infected brown rats that are found in virtually every corner of the world, living in close association with humans, yet the incidence of SEOV human infections varies widely between Asia, where its pathogenic potential is well recognized, and the remainder of the world, where very few human cases have been documented. The even greater exposure to SEOV-infected pet rats, potentially likewise on a global scale, might significantly change this situation.

SEOV infections should be considered in patients presenting with a febrile illness associated with transient thrombocytopenia and proteinuria, accompanied by rapidly resolving hepatitis, especially if a history of possible exposure to brown rats is elicited.

## Figures and Tables

**Figure 1 viruses-11-00652-f001:**
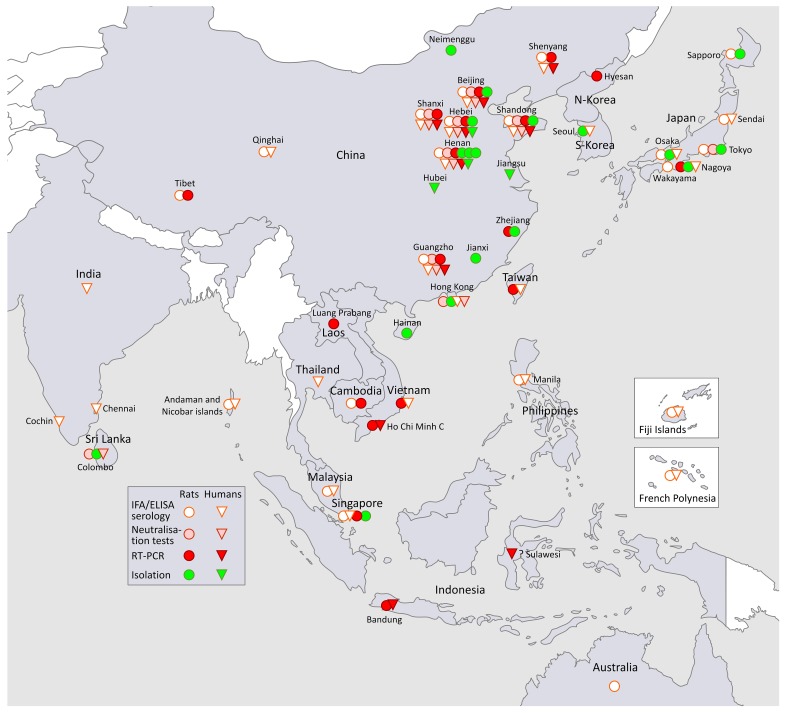
The map does not aspire to completeness and refers only to data given in the text. The first SEOV isolation occurred in 1980 in S. Korea [8], followed in 1982 by China [20], and in 1983 by Japan [21]. Whereas China suffered mixed HTNV-SEOV epidemics, involving thousands of cases, each year, laboratory rat-transmitted SEOV outbreaks were an extensive and recurrent problem in several Japanese medical institutions, during the 1970–1980s (see text). The question mark in Sulawesi, Indonesia, refers to a SEOV infection in a German tourist [22], which may, however, have occurred after his return home (see text under “Germany).

**Figure 2 viruses-11-00652-f002:**
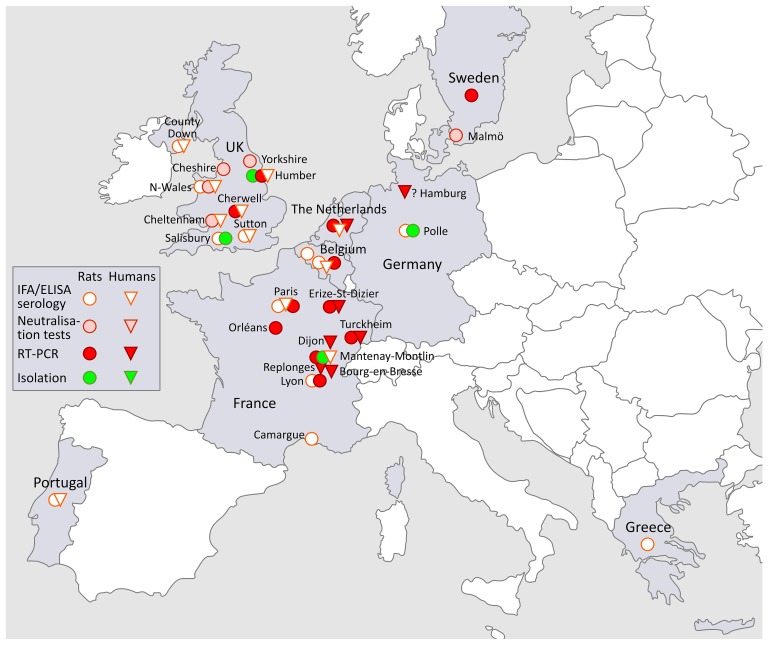
The map does not aspire to completeness and refers only to data given in the text. Except for IR461, isolated in 1984 in Porton Down, Salisbury, UK [90], and for the Polle 18 SEOV strain, isolated in 1988 in Germany (Lower Saxony) [91], all other European SEOV isolates are of more recent dates (see text). The first (1978) western human SEOV infections, propagated by laboratory rats, were clinically described in Belgium [92,93], whereas the earliest series of 16 clinically and serologically proven SEOV-HFRS cases, after exposure to wild rats, was documented in County Down, Northern Ireland, UK, 1994 [94]. For the 1983 and 1988 probable isolated SEOV-HFRS cases in Glasgow and elsewhere in the UK, see the Supplementary Table S1 in the review by McElhinney et al. [95].

**Figure 3 viruses-11-00652-f003:**
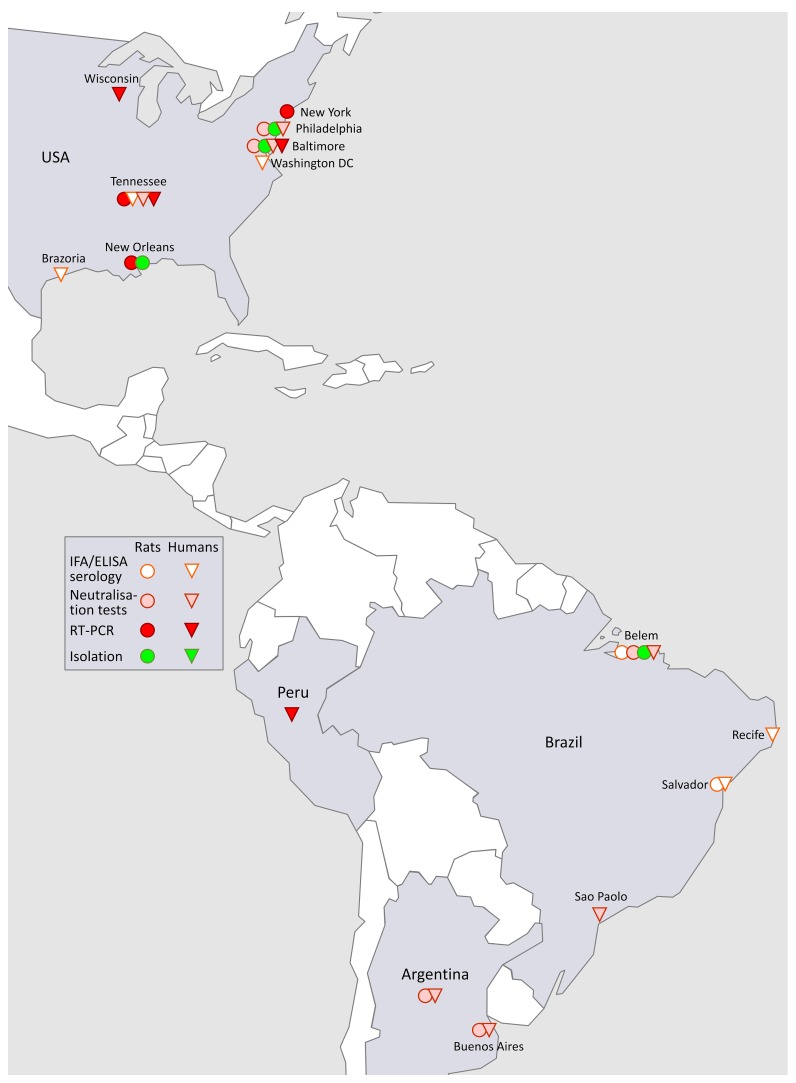
The map does not aspire to completeness and refers only to data given in the text. The first SEOV isolation from a wharf rat captured in Philadelphia, USA, succeeded in 1984 [9], and was rapidly followed by other isolations from wharf rats in different port cities in North and South America (Belem, Brazil) (see text). The earliest (January 2, 1993) description of eight clinically and serologically documented SEOV-HFRS cases, occurring after local floods in 1990, came from Recife, Brazil [123]. They also constituted the first proof of hantaviral pathogenicity for humans in the New World.

**Figure 4 viruses-11-00652-f004:**
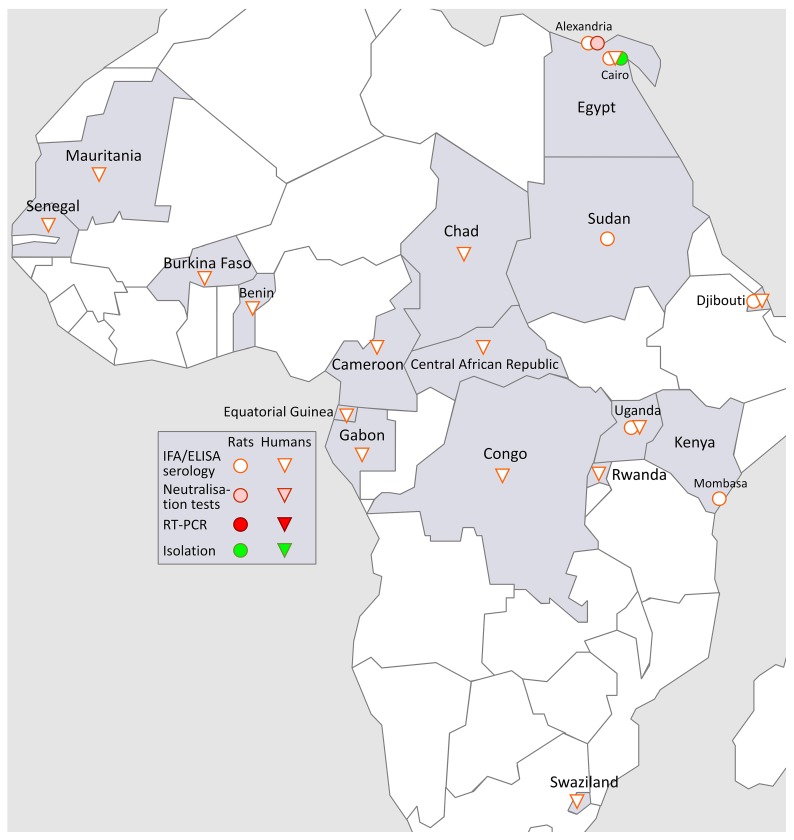
The map does not aspire to completeness and refers only to data given in the text. The first and only SEOV isolations in Africa to date were obtained from wharf rats in Cairo, Egypt, in 1983–1984 [39]. Despite strong evidence since the 1980s for the presence of a SEOV-like agent in humans and/or wild rats in 17 different African countries, no full clinical and serological description of an autochthonous SEOV-HFRS is available so far (see text).

**Table 1 viruses-11-00652-t001:** Characteristics of six reported (1994–2012) SEOV-HFRS cases in the USA, after reported or supposed exposure to wild rats.

Case no.	Reference	Location USA	Peak S. Creat.	Peak Proteinuria	Nadir Platelets	Transaminitis	Lab Confirmation	ICU Treatment	Special Features
1	1994, Glass [12]	Baltimore, MD	3 mg/dL	1700 mg/24hrs	N.R.	+++	PRNT	N.R.	Dyspnea and pleural effusions
2	1994, Glass [12]	Baltimore, MD	N.R.	954 mg/24hrs	N.R.	+	PRNT	N.R.	Dyspnea
3	1994, Glass [12]	Baltimore, MD	1.4 mg/dL	550 mg/24hrs	N.R.	+	PRNT	N.R.	Pleural effusions and hemoptysis
4	2009, Woods [151]	Baltimore, MD	12.3 mg/dL	600 mg/dL 3+	36,000/mm^3^	+++	RT-PCR	6 x Dialysis	Cough, high CK, myoglobulinuria
5	2010, Nielsen [152]	Milwaukee WI	1.53 mg/dL	2 +	26,000/mm^3^	+++	RT-PCR	none	Hypoxia, pleural effusions, ferritin 38,394 ng/mL. Liver biopsy
6	2012, Roig [153]	Brazoria, TX	6 mg/dL	2 +	114,000/mm^3^	++	ELISA	Ventilation	Fatal ARDS

NR: not registered. PRNT: plaque reduction neutralization test. RT-PCR: reverse transcription-polymerase chain reaction. ARDS: acute respiratory distress syndrome. The very high level of ferritin disclosed in case 5 [152] was strongly indicative of an underlying secondary hemophagocytic lymphohistiocytosis (HLH), but was not discussed by the authors. The liver biopsy appeared non-contributive for diagnosis. The severity degree of accompanying, but rapidly self-remitting liver inflammation, was expressed as + to +++, according to the maximum reported levels of hepatic transaminases.

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
