# Peer review of "Wild Rats, Laboratory Rats, Pet Rats: Global Seoul Hantavirus Disease Revisited"

_viruses, 2019, doi:10.3390/v11070652_

Reviewer 1 Report

This is by far the best and most comprehensive review of Seoul virus infection of rats from around the world. Excellent collaborative work by a diverse group of authors. I like the novel figures developed.

My comments are only minor (and few):

Define HCPS on line 69 (first use)

Be consistent with use of Seoul orthohantavirus throughout (e.g., line 21)

A paper published earlier this year made the strong case for the role of pets and pet stores as sources of exposures to rodent-borne viruses, including Seoul virus--this seems like a very relevant paper to cite (Childs et al. 2019 Front. Ecol. Evol. doi.org/10.3389/fevo.2019.00035)

Author Response

Thank you for the compliments on the final result of our common work.

On request of reviewer 2, a succinct general Introduction on hantaviruses has been added, in which the abbreviations “HFRS “ and “HCPS” have been explained, permitting their non-explained use further in the text.

Likewise, the use of the term “ Seoul orthohantavirus” was used in the current Section 2 “Prelude”, with its abbreviation “SEOV”, as used further in the text.

Thank you also for the very welcome suggestion to use the recent excellent paper by JE Childs et al. 2019 FEE, which has now been cited indeed several times in the final version of the text, to be submitted again to Viruses.

Reviewer 2 Report

In their manuscript Clement and co-authors review the literature on Seoul orthohantavirus. The  language of the manuscript is very clear and easy to read, however, the "story" of the manuscript could -in my opinion- flow better. At times the authors seem to adopt a "besserwisser" attitude, which is hopefully not intentional.

The introduction" begins with a few SEOV case descriptions, which is rather unexpected. In overall the introduction seems more like a historical overview on the early days of hantavirology (which is an interesting read from these experienced authors) rather than true introduction to the topic. I would suggest the authors to write a separate short introduction on hantaviruses first, and label the current introduction something else. Or if the authors like to have the piece called introduction, I would suggest starting with a paragraph (or two) of more general introduction to hantaviruses. Also it might be beneficial to briefly describe the course and symptoms of hantavirus infection in man.

Is "1). SEOV in the Far East and South-East Asia (Figure 1, Asia)" on line 103 actually a heading, does the introduction continue 104=>? These bits could be divided in sections under different headings, if the authors intend to describe SEOV occurrences in different countries/regions.

At times the authors jump between years (naturally this cannot be always avoided), which confuses the reader at times. Could the authors try to present (most of) the data in chronological order? The authors could consider collecting the findings of SEOV in different species in to a table, that could be a valuable resource for other researches of hantavirus field.

Author Response

Following your suggestion, a separate short Introduction on hantaviruses in general has been added, followed now by the current Section 2 “Prelude”. The reasons for mentioning in this “Prelude” the recent American pet rat-transmitted SEOV-HFRS cases, and the further better structured divisions in different following (sub)sections are briefly explained in the track changes of the final version in Word format, also submitted to Viruses.

We attempted to better arrange in chronological order the major points made in the text. A brief explanation for some “chronological jumps” in the text is included in the track changes. We adjusted the order and numbering of references accordingly to reflect these changes.

We considered the suggestion to prepare a Table of different species found infected with SEOV, but concluded that there is little or no evidence of a role of the different host species in the epidemiology of SEOV infections, and that most strains detected are nearly identical to those found in Rattus norvegicus. Moreover, the only regular SEOV rodent reservoir with a worldwide spread remains Rattus norvegicus, all other incidental rodent hosts being restricted to the (Far) East.

Finally, we apologize if the manuscript reflected a ”besserwisser attitude. “ We assure the Reviewer that such an attitude was not intentional, and we have reviewed the text in an attempt to eliminate such indications.